# Open-source microscope add-on for structured illumination microscopy

Mélanie T. M. Hannebelle[1,2,6,7], Esther Raeth [1,2,7], Samuel M. Leitao [1], Tomáš Lukeš[1], Jakub Pospíšil [3,4], Chiara Toniolo [2], Olivier F. Venzin [2], Antonius Chrisnandy [2], Prabhu P. Swain[1], Nathan Ronceray [1], Matthias P. Lütolf [2], Andrew C. Oates [2], Guy M. Hagen [5], Theo Lasser [1], Aleksandra Radenovic [1], John D. McKinney [2] & Georg E. Fantner [1] ✉

Super-resolution techniques expand the abilities of researchers who have the knowledge and resources to either build or purchase a system. This excludes the part of the research community without these capabilities. Here we introduce the openSIM add-on to upgrade existing optical microscopes to Structured Illumination super-resolution Microscopes (SIM). The open-SIM is an open-hardware system, designed and documented to be easily duplicated by other laboratories, making super-resolution modality accessible to facilitate innovative research. The add-on approach gives a performance improvement for pre-existing lab equipment without the need to build a completely new system.

The complexity and cost of a scientific instrument have a large impact on the number of scientific teams with access to the instrument for their research[1,2] and thus to which extent a microscope technique will contribute to making scientific discoveries.

Super-resolution microscopy has become a powerful tool enabling researchers to study biological processes with an exceptional resolution by overcoming the diffraction limit of light inherent in traditional imaging methods. Structured illumination microscopy[3] (SIM) is a super-resolution technique particularly suited for biological applications due to reduced phototoxicity and high-resolution[4–6]. SIM increases microscopy resolution by using a spatially structured pattern to illuminate the sample, in combination with image reconstruction to deduce information that would normally be unresolved. Although it provides a moderate increase in resolution compared to other super-resolution techniques such as STED[7] or SMLM[8,9] (two-fold maximum increase in lateral resolution for linear SIM[3]), biological imaging benefits from higher viability when utilizing SIM as it requires lower light excitation intensity resulting in reduced phototoxicity and photo-bleaching compared to other techniques[1].

The hurdle for most research groups is the limited access to super-resolution microscopes, which forces them to rely on lower-resolution conventional fluorescence microscopes instead. To promote technology dissemination among researchers, open-source microscopy initiatives have an significant impact on the distribution of optical designs and instrument building techniques[10,11]. Several initiatives have introduced stand-alone research-grade open-source microscopes, among them the light-sheet microscope openSPIM[10,12], the wide-field microscope OpenFlexure[13], the 3D-printed optical toolbox UC2[14], the AttoBright[2] for single-molecule detection, and the miCube[15] for single molecule localization microscopy. Although open-source microscopes have provided scientists with an easier access to super-resolution imaging methods, there is a lack of open-source integrative systems. The absence of integrative open-source hardware systems presents a challenge within the scientific community and frequently results in the redundant development of devices across various laboratories, which not only consumes valuable time and resources but also hinders the potential for collaborative innovation. Many laboratories already have conventional

[1]School of Engineering, Swiss Federal Institute of Technology (EPFL), Lausanne, Switzerland. [2]School of Life Sciences, Swiss Federal Institute of Technology (EPFL), Lausanne, Switzerland. [3]Faculty of Electrical Engineering, Czech Technical University in Prague, Prague, Czech Republic. [4]Department of Medical Biology, UiT The Arctic University of Norway, Tromsø, Norway. [5]BioFrontiers Center, University of Colorado Colorado Springs, Colorado Springs, CO, USA. [6]Present address: Center for Innovation in Global Health, Stanford University, Stanford, CA, USA. [7]These authors contributed equally: Mélanie T. M. Hannebelle, Esther Raeth. ✉e-mail: georg.fantner@epfl.ch

fluorescence microscopes readily available. Thus, appending existing commercial microscopes using open-source integrative systems would facilitate the usage of super-resolution techniques.

## Results

Here we introduce the openSIM, a microscope add-on that extends the imaging capabilities of already existing microscopes without the need to build a completely new system. At the same time, this add-on does not impact the function of additional instrumentation connected to the microscope, such as incubation chambers to control temperature, $CO_2$ and humidity, microscope stages, syringe pumps, microfluidic systems etc. depending on the samples being studied. We believe that this reduces the entry barrier to adopt open-source super-resolution technology.

As an open-source microscope, we provide detailed documentation and instructions[16] enabling other scientists to build their own openSIM add-on (Supplementary Fig. 1, Supplementary Fig. 2) to upgrade their microscope setups into super-resolution instruments, and benefit from improved image datasets for their biological research. The assembled openSIM add-on is a compact module (Fig. 1a) that connects to the illumination port of standard fluorescence microscopes (Fig. 1b, Supplementary Fig. 3). It replaces the illumination source of the microscope and provides the striped illumination pattern required for SIM imaging (Supplementary Fig. 4). For this, collimated light from high-power LEDs illuminates the liquid-crystal-on-silicon (LCOS) spatial light modulator (SLM), which generates the pattern to be projected on the sample via the openSIM tube lens and the microscope objective (Fig. 1c, Fig. 1d). LCOS displays are widely used as SLM for SIM, offering high data acquisition rates, flexibility and are readily available at moderate cost. Ferro-electric LCOS (FLCOS) were previously used for 2D SIM[17–19] and 3D SIM[3,5,20]. A software interface brings together on-the-fly pattern control, illumination color control, light intensity control, camera control (exposure time, camera gains, data saving etc.), and closed loop control of the openSIM heat sink temperature (Supplementary Fig. 5). An interface box containing a DAQ (Data Acquisition) device and connectors allows simple cabling between the elements of the system (openSIM add-on, camera, computer and optionally microscope z-stage for automated volumetric imaging) (Fig. 1a, Supplementary Fig. 6). The openSIM software guides the user to tune the optimal pattern size, depending on the chosen microscope objective. The images acquired with patterned illumination are saved in a format directly compatible with SIMToolbox[21], an open-source MATLAB based software for SIM image reconstruction.

To evaluate the performance of the openSIM, we imaged 100 nm fluorescent beads (Supplementary Fig. 7, Supplementary Fig. 8). After SIM reconstruction, the width of the point spread function was 169 nm (average value on 20 beads), compared to 294 nm for wide-field illumination (Fig. 2a). Beads in close proximity that were not resolved with wide field microscopy were resolved with openSIM (Fig. 2b): while one single large peak was visible on the wide-field data, two distinct peaks were observed on the SIM image. In addition to increasing the lateral resolution of the microscope, SIM provides an axial optical sectioning effect, more precisely, it reduces the contribution of out-of-focus light to the image. In Fig. 2c, we compare a wide-field image and an openSIM image of a fixed mouse intestinal organoid. While out-of-focus cells create a blurred background and lower the contrast in the wide-field image, the openSIM image only contains contributions from cells in the focal plane, thereby greatly increasing the image quality.

To demonstrate the usage of openSIM across a wide spectrum of biological applications, we selected a range of biological samples encompassing large-scale biological specimens as well as samples at cellular and subcellular levels. We imaged a mouse intestinal organoid (Fig. 1e), a valuable tool for studying intestinal diseases and interactions between the epithelial tissue, immune system and microbiome. The openSIM add-on also permits studying whole model organisms, such as zebrafish embryos (Fig. 2d). In both samples, the advantages of the optical sectioning are particularly evident. The openSIM was also employed to examine structures at a cellular and subcellular scale, such as the distribution and organization of individual *Mycobacterium smegmatis* bacteria within macrophages during infection (Fig. 2e). Tubulin was imaged on a fixed pulmonary artery endothelial cell, using the blue excitation channel of the openSIM (Supplementary Fig. 9, Supplementary Fig. 11); this shows the benefit of the openSIM add-on for studying mechanobiology at the subcellular scale. We also mounted the openSIM add-on to a scanning ion-conductance microscope (SICM)[22], which is a scanning probe microscopy (SPM) technique, for correlated imaging of the cell surface with SICM and the cytoskeleton with the openSIM (Fig. 2f). This application further highlights the versatility of the openSIM add-on, showcasing its ability to seamlessly integrate with complex instruments without compromising their performance.

## Discussion

In addition to the two main factors limiting the quality of microscopy images - diffraction of light and limited photon budget - the complexity and cost of an instrument significantly impact the extent to which a microscopy technique will contribute to scientific discoveries. When designing the openSIM add-on, our intention was not to compete with the most advanced SIM systems. Instead, our focus was on enhancing existing instruments with minimal modification and cost-effective components. The main difference between the optical design of openSIM and other conventional systems is the excitation pattern generation approach and the use of an incoherent light source. Conventional SIM systems use high-frequency fringe patterns formed by interference of diffraction beams in the sample plane to illuminate the sample[5,17–20,23]. The approach for the illumination pattern formation of openSIM follows a non-diffraction or interference-based method using non-coherent light, described in[21,24–27], in which an image of the SLM is formed directly in the sample plane instead of being created by interference of diffraction beams.

One of the primary benefits of this approach, as opposed to systems relying on interference, lies in the inherent simplicity of the optical setup with a small number of basic optical components and reduced demands for precise positioning and alignment of these optical elements, making the overall system less prone to alignment-related challenges. For SIM systems in which the illumination pattern is generated based on interference, it is important to block undesired diffraction orders[17,28] and to maintain the polarization state[5,29]. This requires additional optical components[5,17,18,29] for which precise design and positioning are crucial due to wavelength and pattern orientation considerations. In contrast, SIM systems based on direct imaging of the pattern into the sample plane, do not require these components, which reduces the complexity of the optical design and increases the flexibility by avoiding wavelength- or pattern orientation constrains. This enables an uncomplicated switching between multiple wavelengths and the use of different pattern orientations.

Despite the described advantages of the chosen optical design, it presents certain drawbacks that potentially result in a reduced performance. Placing the microdisplay in the image plane for direct pattern imaging into the sample plane results in an inefficient use of the illumination light. Nonetheless, the employed high-power LEDs possess sufficient brightness to offset this inefficiency[24]. Furthermore, polarizing elements such as the LCOS display and the PBS cube exhibit suboptimal polarization properties leading to a decrease of the pattern contrast[30].

A main factor influencing the resolving performance of SIM is the highest achievable illumination pattern contrast. Non-coherent systems[24–27,31] such as openSIM, which employ incoherent light from high power LEDs, exhibit a reduced maximal attainable contrast. This consequently leads to a diminished enhancement in resolution compared to coherent systems[3,5,17–20,23] utilizing coherent laser light.

For coherent illumination SIM, the coherent optical transfer function applies, where the pattern contrast remains constant with increasing spatial frequency. In contrast for incoherent illumination SIM, the incoherent optical transfer function applies, causing pattern contrast to decrease as spatial frequency increases[32], leading to a diminished contrast for high-frequency patterns. Nevertheless, despite the potential higher achievable resolution of coherent illumination SIM, the use of coherent light sources comes with several drawbacks. Lasers

are expensive, spacious and only available for certain wavelengths. Since we were aiming to provide an inexpensive and compact system which is straightforward to reconstruct, aspects such as cost-effectiveness, availability, small dimensions and safety of the used components were of main importance. LEDs are a cost-effective, simple to handle and compact option, offering versatility in available wavelengths and simplifying the integration of multiple excitation channels and a simple exchange. Moreover, in contrast to LEDs which exhibit a

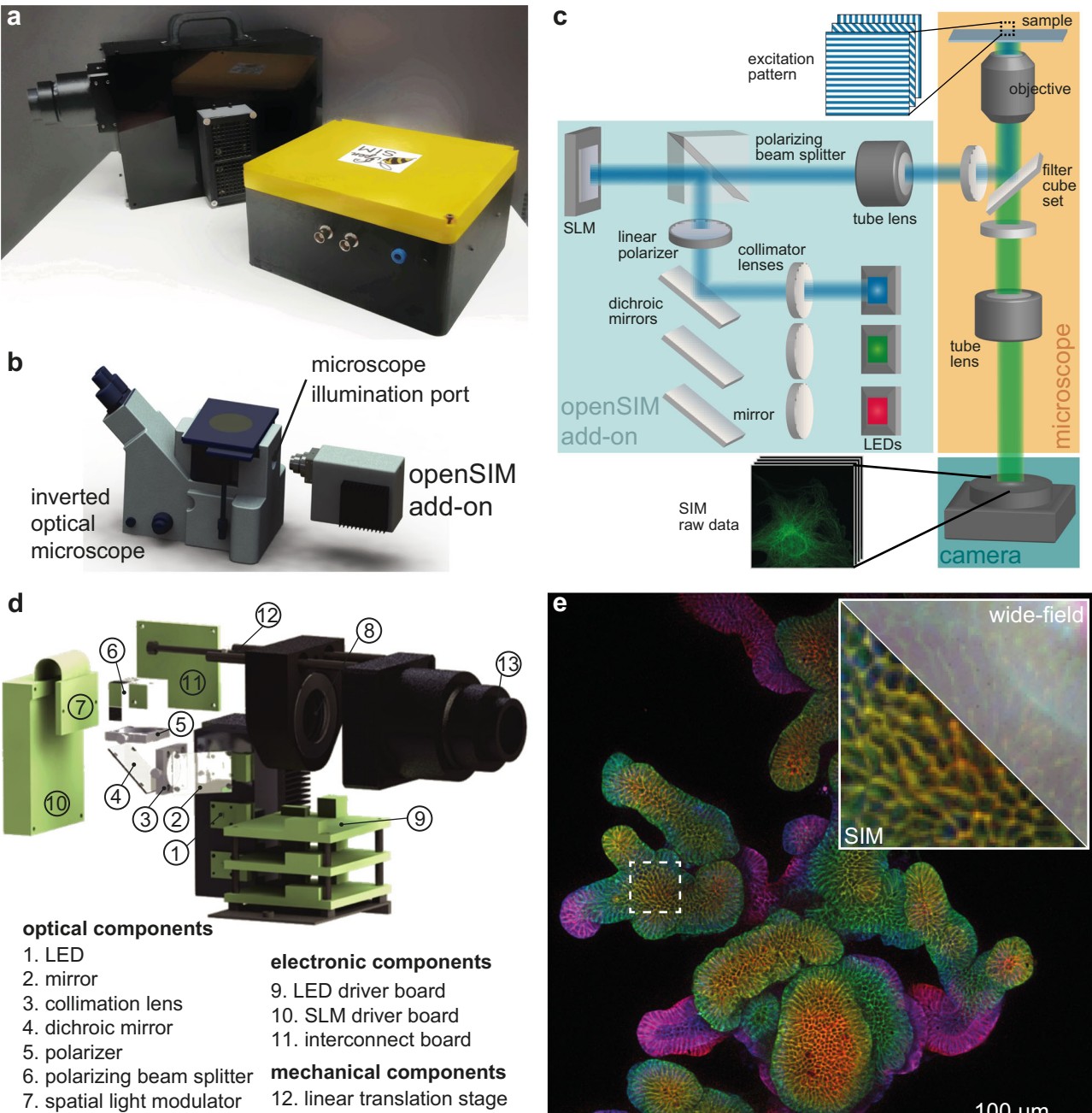

**optical components**
1. LED
2. mirror
3. collimation lens
4. dichroic mirror
5. polarizer
6. polarizing beam splitter
7. spatial light modulator
8. tube lens

**electronic components**
9. LED driver board
10. SLM driver board
11. interconnect board

**mechanical components**
12. linear translation stage
13. illumination port adapter

**Fig. 1 | Design and assembly of the openSIM microscope add-on. a** Photo of the openSIM add-on (back) and its interface box (front). **b** 3D rendering illustrating the connection between the openSIM and a standard inverted optical microscope. **c** Schematic of the optical design of the openSIM add-on. LEDs are collimated to illuminate the spatial light modulator (SLM). The SLM generates a pattern by changing the polarity of the reflected light. The illumination pattern is transferred into the infinity space with a tube lens. **d** 3D rendering of the main optical, electrical and mechanical components of the openSIM. For simplicity only the components associated with the blue illumination channel are represented. **e** Illustration of the increased resolution and optical sectioning when using openSIM compared to wide-field imaging. Whole mount 3D color coded image of fixed mouse intestinal organoids labeled for E-cadherin (epithelial cell junctions). The inset represents a zoom in the area highlighted with a dotted square. Part of the image is not processed (equivalent wide-field image, top right corner), while the bottom left corner is a SIM reconstruction.

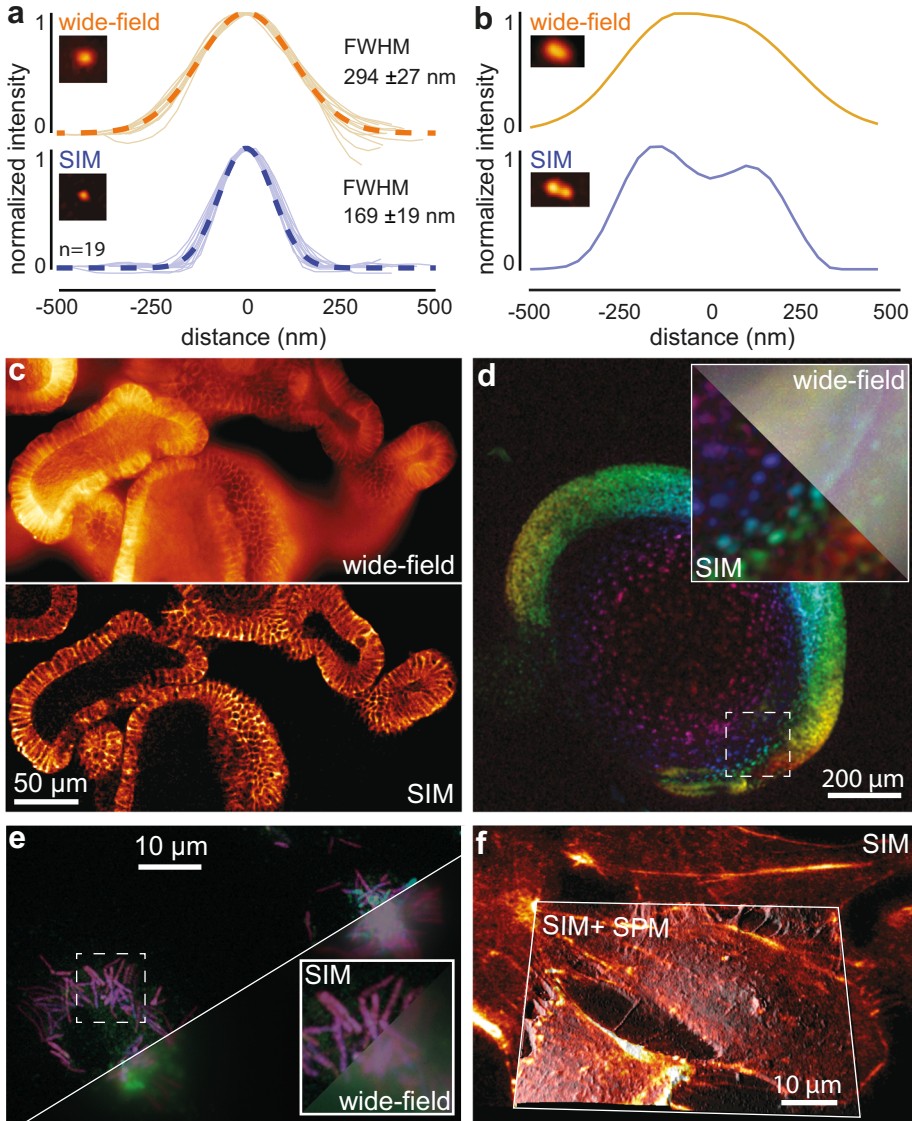

**Fig. 2 | openSIM increases resolution and optical sectioning compared to wide-field illumination. a** Comparison between the point spread function of 20 isolated 100 nm beads, with wide-field illumination (top) and with the openSIM (bottom), with blue illumination. FWHM: full width at half maximum. Dotted line: average Gaussian fit. The picture inset shows a representative 100 nm bead. **b** Comparison between the openSIM and wide-field illumination to resolve two individual beads placed close to each other. **c** Illustration of the optical sectioning effect when using the openSIM. On the wide-field image, out of focus light is noticeable in the image; on the openSIM image, only the part of the sample directly in focus is visible. Sample: fixed mouse intestinal organoids labeled for E-cadherin. **d** 3D color coded image of a live zebrafish embryo with cell nucleus staining. The inset represents a zoom in the area highlighted with a dotted square. Part of the image is not processed (equivalent wide-field image, top right corner), while the bottom left corner is a SIM reconstruction. **e** 3D color-coded image of fixed macrophages infected with *Mycobacterium smegmatis* bacteria. **f** Combined SIM and scanning probe microscopy (SPM) image of fixed actin-stained COS-7 cells. The SIM image is a maximum intensity projection, overlaid as a skin on the topography image obtained with a scanning ion conductance microscope.

low spatial coherence, the high lateral coherence of laser light can be problematic due to stray interference patterns[28].

Overall, the openSIM design has the advantage of providing a simple optical design with reliable performance, in addition to lower damage due to phototoxicity and bleaching while requiring minimal alignment and maintenance. The openSIM quantitatively improves the quality of microscopy images (Fig. 2), both in terms of resolution (Fig. 2a, b) and optical sectioning (Fig. 2d). Improving the capabilities of existing research grade instruments through the integrative open-hardware add-on approach allows researchers to increase the performance of their instruments for a cost less than one order of magnitude lower compared with commercial SIM microscopes. The openSIM upgrades these dedicated and customized microscopes without reducing their original capabilities, since the setup remains compatible with the specialized equipment often required for studying biological organisms. We aimed for an add-on which facilitates an effortless integration with various commercially available microscopes (such as Olympus, Zeiss, Leica) through the straightforward replacement of the tube lens and illumination port adapter. However, it's worth considering that the microscope-specific components may limit deployability. To potentially enhance versatility, manufacturers might consider promoting standardized interfaces and sharing blueprints. This could offer laboratories a valuable resource for constructing and tailoring hardware components which could simplify the setup and streamline workflows for researchers. Consequently, this might encourage more laboratories to openly share their work, foster collaborative adaptations, and lessen the barriers that could impede other labs from adopting open-source devices developed by different labs

As the openSIM is a versatile tool with a well-documented assembly, simple maintenance, and linked to an open-source community, we believe that it has the potential to make super-resolution microscopy not only a cutting-edge technique but also a daily tool for biological research.

## Methods

The optical, mechanical and electronic parts of the openSIM are depicted in Supplementary Fig. 1. The estimated cost for all components necessary to build up an openSIM add-on is around 10.000 €. A detailed documentation, assembly and user instructions can be found on our wiki-page[16] (https://opensim.notion.site/).

### Optical design of openSIM

For SIM systems in which the illumination pattern is generated based on interference, commonly LCOS SLMs in phase-only mode are used. Coherent light is collimated onto the SLM, which serves as a phase grating, leading to the diffraction beams. The illumination pattern is then created by the interference of two diffraction beams (for 2D SIM) or three diffraction beams (for 3D SIM) in the sample plane.

In contrast for SIM system based on direct imaging of pattern into the sample plane, the LCOS SLM is placed in the primary image plane of the microscope and used in combination with a polarizing beam splitter (PBS) in amplitude mode to serve as a computer-controlled binary mask. The microdisplay is imaged onto the sample via a polarizing beam splitter, an additional tube lens and the microscope's objective.

LCOS microdisplays used in amplitude mode operate as an array of individually addressable, electrically switchable quarter-wave plates, formed by the ferroelectric liquid crystal pixels with a reflective backing. By manipulating the polarization of the incident light in a specific manner through the SLM, the combined use of the SLM and a polarizing beam splitter permits the transmission of light from selected pixels while blocking others. Whereas active pixels function as quarter-waveplates and thus alter the polarization of the incoming light, inactive pixels can be considered as mirrors which reflect the incoming light without changing the polarization. Linearly vertically polarized illumination light is directed onto the display using a polarizing beam splitter (PBS) cube. When vertically polarized light travels through an active pixel and back it becomes horizontally polarized light. Only light which is horizontally polarized, corresponding to the light from the active pixel, is transmitted by the PBS, creating a binary illumination pattern. The openSIM add-on can also provide homogeneous illumination for wide-field imaging, acting as a conventional 3-color light source if SIM is not needed

LCOS are most commonly used for SLM-based pattern generation due to their high level of performance. The use of DMDs for SIM[33–36] would be an alternative. They are more cost-effective and the implementation of the timing scheme is less difficult. Conversely, LCOS microdisplays exhibit reduced diffractive efficiency, resulting in fewer diffraction losses into higher orders that may not be captured by the imaging lens[24].

Depending on the optical configuration, the spatial frequency of the illumination patterns employed and the applied reconstruction algorithm, SIM can be used for optical sectioning by reducing out of focus light (OS-SIM)[28,37] and for super-resolution imaging by enhancing the resolution (SR-SIM)[3,31]. For OS-SIM relatively coarse illumination patterns are used[38], while SR-SIM utilizes illumination patterns with a high spatial frequency close to the diffraction limit[25]. LCOS microdisplays provide the flexibility to display illumination patterns with arbitrary spatial frequencies.

### Optical components for the openSIM

Three high power PT54 LEDs (PT-54-TE, Luminus Devices; dominant wavelengths: red-amber $\lambda_d$ = 613 nm, green $\lambda_d$ = 525 nm, blue $\lambda_d$ = 460 nm, emitting area of 5.4 mm²) are used to generate the illumination light. The light emitted by the three LEDs is deflected by three mirrors (#43-875, Edmund optics), is collimated using 50 mm lenses (#66-018, Edmund optics), and combined into one optical path with dichroic mirrors (#69-898, #69-900, Edmund optics). A polarizer selects for the specific polarization reflected by the polarizing beam splitter (#49-002, Edmund optics). Then, the light reaches the spatial light modulator (QXGA-3DM-STR, ForthDimension Displays). A tube lens (TTL180-A, Thorlabs), compatible with the commercial microscope being used in this paper (IX71 and IX81, Olympus), is placed in such a way that the SLM is in the back focal place of that tube lens. Commercially available connectors (SM2Y1, LCP11, Thorlabs) were used to connect the openSIM to the illumination port of the microscope. The openSIM is designed to be easily adaptable to other commercially available microscopes (e.g. Olympus, Zeiss, Leica) by changing the tube lens and the illumination port adapter which are available in the Thorlabs catalog.

### Mechanical components for the openSIM

The openSIM was designed with Solidworks software and was printed with a consumer grade 3D printer with black PLA as a printing material. We used a 3 mm thick black acrylic sheet for the panels of the enclosure. The sheets were trimmed to the corresponding size with a laser cutter. The position of the tube lens is adjusted using a linear translation stage. The translation stage was built with two linear shafts (SSFJ6-100, Misumi), a precision feed screw (XBRF6, Misumi) and two wire springs (WF8-40).

### Electronic components for the openSIM

The LEDs are powered and controlled using the boards and connectors supplied in the DK114N3 development kit (Luminus devices). The kit consists of high current driver boards and cable assemblies for each channel to provide the LED drive current for high brightness. A custom interconnect board was designed to facilitate the connection between the elements of the openSIM, such as the data acquisition device, LEDs, thermistor and fan for temperature control, SLM and camera.

The openSIM interface box allows a simple cabling between the elements of the system (openSIM add-on, camera, computer and optionally z-actuator). It contains a DAQ (USB-6000, National Instruments), two power supplies (RD-35A, Mean Well; ZWS300BAF-12, TDK) and several connectors to interface the different components. Several analog and digital input and output lines of the DAQ remain available for customization by users.

### LabView interface and openSIM control

We have designed a complete instrument interface with LabView, which brings together on-the-fly pattern control, illumination color control, light intensity control and closed loop temperature control of the LED heat sink (Supplementary Fig. 5). It also saves the detailed acquisition parameters and pattern sequence used during the respective image acquisition. At the moment we offer several software versions with different functionalities in order to provide an adequate option for user with diverse requirements. For users who want to modify or customize the software we provide the full LabView project including camera control (exposure time, camera gains, binning options) and data handling (pre-formatting the data to be compatible directly with the open-source SIMToolbox SIM processing algorithm). The first version includes a full camera control for the camera models Andor Zyla PLUS sCMOS (6.5 μm pixel size) and Andor iXon3 EMCCD (8 μm pixel size). The modular design of the software provides the possibility to include the control of other commercial cameras. For user who want to primarily use the software, we offer an application (exe) of the openSIM software which is not camera model specific and does not require costly software licenses.

## Synchronization camera, SLM and LEDs

SIM images can exhibit artifacts associated with the illumination pattern. Thus, the timing of the system's sub-components, namely camera, SLM, LED, has to be precisely synchronized (Supplementary Fig. 10). The camera's exposure signal is employed to initiate the sequential projection of the patterns chosen within the pattern sequence. LCOS microdisplays cannot continuously display patterns since each pixel state must be inverted after every image to avoid a charge build-up. To facilitate this, the illumination source must be temporarily disabled during the refresh cycles, necessitating a precisely synchronized timing scheme for both the microdisplay and LEDs. The time period, which delineates the intervals for displaying the pattern or the inverted pattern, is determined by predefined timing schemes provided by the manufacturer of the SLM (ForthDimension).

## Illumination pattern

In order to obtain nearly isotropic resolution improvement in 2D linear SIM, the periodic pattern needs to be shifted through three phases (120° to each other) and for each of three orientations (0°, 60°, 120°), yielding nine raw images[3]. In our experiments, a sequence of periodic line patterns with 3 orientation angles and 4 images per angle (12 different patterns in total) was used, in total resulting in a homogenous illumination. The chosen illumination patterns had a high spatial frequency. The set of patterns was designed according to Lukeš et al.[25] so that the pattern for different orientation angles have a similar mark-to-area ratio (MAR) (i.e. fraction of the illumination area)[28], and spatial frequency. Because the pixels on the microdisplay have a square shape, additional patterns were used in the diagonal directions to ensure even coverage across the entire image while maintaining a roughly consistent spatial frequency for the patterns[25]. While three angles are sufficient for SIM image reconstruction, using more angles can potentially increase the spatial resolution and improve the robustness and accuracy of the image reconstruction. Nevertheless, this comes at the expense of extended acquisition times, increased photobleaching and computational cost. We thus used a pattern repertoire with an additional angle orientation (4 angles and 3 or 4 images per angle, 14 different patterns in total) for non-biological or fixed samples where photobleaching is not of importance.

## Sample preparation

The organoids were derived from Lgr5–eGFP–ires–CreERT2 mouse intestinal crypts and grown in 3D Matrigel culture under expansion conditions (ENRCV)[39]. The preparation of the imaging sample was adapted from Gjorevski et al.[40]. In short, the organoids were fixed with 4% paraformaldehyde (PFA) in PBS 1X. The fixed samples were permeabilized with 0.2% Triton X-100 in PBS 1X (1 h, room temperature) and were blocked with 0.01% Triton X-100 in PBS 1X containing 10% goat serum (3 h, room temperature). The organoids were then stained using a monoclonal anti-E cadherin antibody (ab 11512; Abcam) followed by secondary antibody Goat-α-Rat Alexa Fluor 568 (A-11077; ThermoFisher Scientific). Extensive washing steps were performed subsequent to each antibody incubation step. The imaging slide was prepared by mounting the stained organoids on glass coverslips with addition of mounting medium Fluoromount-G® (0100-01; SouthernBiotech).

Bovine pulmonary artery endothelial (BPAE) cells with stained F-actin and tubulin were acquired from Thermofisher (FluoCells prepared slide #2). F-Actin is labeled with Texas RED-X phalloidin, and tubulin is labeled with anti-bovine α-tubulin mouse monoclonal 236-10501 conjugated with BODIPY FL goat anti-mouse IgG antibody.

Bone marrow derived macrophages (BMDMs) were differentiated by seeding $10^6$ bone marrow cells from C57BL/6 mice in petri dishes and maintaining them in DMEM supplemented with 10% heat-inactivated fetal bovine serum (HI-FBS), 1% sodium-pyruvate, 1% GlutaMax and 20% L929-cell-conditioned medium as a source of granulocyte/macrophage colony stimulating factor. After 1 week of cultivation at 37 °C with 5% $CO_2$, the adherent differentiated macrophages were gently detached from the plate using a cell lifter and resuspended in DMEM supplemented with 5% HI-FBS, 1% sodium-pyruvate, 1% GlutaMax and 5% L929-cell-conditioned medium. $10^4$ cells were seeded in a 35 mm cell culture micro-dish with a coverslip bottom (IBIDI) and allowed to adhere for 24 h before infection.

A GFP expressing *Mycobacterium smegmatis* (*Msm*) strain was cultured at 37 °C in Middlebrook 7H9 (Difco) supplemented with 10% ABS, 0.5% glycerol, and 0.02% tylox pol. 1 ml of culture at $OD_{600}$ 0.5 was pelleted, concentrated 5 times in the medium of the BMDMs and filtered with a 5 μm filter to obtain a single cell suspension. BMDMs were infected with the single cell suspension with MOI 1:1 and incubated at 37 °C with 5% $CO_2$. After 4 h the cells were washed extensively with DMEM to eliminate extracellular bacteria and incubated at 37 °C with 5% $CO_2$ for 48 h to allow the infection to proceed. Before imaging, the cells were stained with CellMask™ Orange (Invitrogen) according to the manufacturer's protocol and fixed with 4% formaldehyde in PBS for 30 min at room temperature.

African green monkey kidney fibroblast-like cells (COS-7), purchased from ATCC were grown in DMEM without phenol red medium (Sigma Aldrich), containing 10% fetal bovine serum. The #1.5 cover glass coverslips were cleaned with piranha solution and coated with fibronectin from bovine plasma (0.5 μM/ml). Then the cells were fixed with 4% PFA in 1xPBS (pH 7.4) for 10 min at room temperature and washed twice for 5 min with PBS (pH 7.4). Prior to staining, the fixed cells were incubated with PBS containing 1% BSA for 30 min to reduce nonspecific background. To visualize F-actin, the phalloidin staining solution (Fluor 488 phalloidin ThermoFisher) was placed on the coverslip for 20 min at room temperature.

Zebrafish were maintained according to standard procedures at the EPFL fish facility, which has been accredited by the Service de la Consommation et des Affaires Vétérinaires of the canton of Vaud – Switzerland (VD-H23) and embryos were staged according to Kimmel et al.[41]. Embryos were obtained from outcrosses of Tg (Xla.Eef1a1:H2B-mCherry) to WT fish by natural spawning and raised in fish water (pH 7.8 ± 0.1, conductivity 500 ± 50 μS). For imaging, embryos were de-chorionated and laterally aligned in conical depressions in a pad of 2% low-melting agarose (Sigma) that was cast in a glass-bottom dish (Matek, 35 mm), as in Herrgen et al.[42].

## Image acquisition

The fixed mouse intestinal organoid, the live zebrafish embryo, the macrophages infected with Mycobacterium smegmatis bacteria and the fixed actin-stained COS-7 cells were imaged using a 20x and 100x (1.3 NA, oil immersion) objective. The 100 nm bead sample and the fixed bovine pulmonary artery endothelial cells were imaged with a 100x objective (1.3 NA, oil immersion). For the image acquisition an Andor Zyla 4.2 Plus camera (pixel format 1048 × 1048) and an Andor iXon3 camera (pixel format 1024 × 1024) were used. We provide example data sets of the fixed 100 nm bead sample, fixed bovine pulmonary artery endothelial cell sample and the live zebrafish sample on Zenodo[43] including the raw data and the reconstructed data with corresponding information about the used parameter setting during image acquisition and image reconstruction.

## Combined SIM/SICM microscopy

Scanning probe microscopy was performed with a custom-made scanning ion conductance microscope[22] (SICM). The sample was actuated in X and Y by a piezo-stage (Piezosystem Jena TRITOR102SG). The capillary was moved in Z by a home-built actuator, operated in hopping mode. The hopping height was 1 μm at 100 Hz rate. The current setpoint used in the hopping actuation was 99% of the normalized current recorded. Images with 256 × 256 pixels were generated.

## Image processing

The raw patterned images were used to calculate the SIM image using the SIMToolbox[21] open-source software. The program provides OS-SIM and SR-SIM processing algorithms. For the presented images, the SR-SIM reconstruction method was used. Volumetric images were rendered using maximum intensity projections and color-coded projections, using Fiji[44]. Fiji was also used for measuring the point spread function profiles in Fig. 2a. The images have been individually brightness-adjusted for presentation, whereas corresponding widefield and SIM images were adjusted in the same way. For the calculation of the Fourier transforms the Fiji FFT plug-in was used. The SICM image with SIM overlay (Fig. 2f) was rendered using the Fiji plug-in 3D-surface-plot[45].

## Supplementary Information

Supplementary information is available for this paper. The open hardware documentation for the openSIM is available at the following webpage[16]: https://opensim.notion.site/

## Reporting summary

Further information on research design is available in the Nature Portfolio Reporting Summary linked to this article.

## Data availability

The imaging data presented in this paper, the source data underlying the analysis presented in Fig. 2a and example data sets are publicly available in the following repository[43]: https://zenodo.org/records/10067217

## Code availability

The data was acquired with a custom-written LabVIEW software utilizing the following commercial software: LabVIEW 2018 SP1 including DAQmx 18.6, Vision Development Module 2018 SP1, Andor SDK3 for LabVIEW; MetroCon 3.3; Microsoft Visual Studio Express 2013. Different versions of this software are publicly available in the following repository[46]: https://github.com/EstherRaeth/openSIM_LBNI

The image reconstruction was performed with the open-source MATLAB software SIMToolbox 1.0 and 2.0 (https://simtoolbox.github.io/). Fiji was used for image processing and image analysis.

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

## Acknowledgements

This research was funded by the EPFL Open Science Fund, the Swiss Commission for Technology and Innovation (CTI-18330.1; G.E.F) and the European Research Council (ERC-2017-CoG, InCell.; G.E.F), Swiss National Science Foundation (205321_134786 and 205320_152675; G.E.F), from the Commission for Technology and Innovation under CTI (18330.1 PFNM-NM; G.E.F). The authors would like to thank Adrien Descloux, David Nguyen and Nicolas Bichon for their valuable contributions.

## Author contributions

M.T.M.H. and G.E.F. conceptualized the idea, M.T.M.H., E.R. designed the instrument, developed the hardware components and the acquisition software, M.T.M.H designed the experiments, E.R. created the project documentation, M.T.M.H., E.R., S.M.L, C.T., O.V., A.C., N.R., prepared the samples and acquired the data, M.T.M.H., E.R., S.M.L analyzed the data, T.L., J.P., P.P.S supported the image reconstruction, M.T.M.H., E.R., S.M.L and G.E.F. wrote the manuscript. G.E.F supervised, conceived, planned the project and provided funding. P.P.S, M.P.L., A.C.O, G.M.H, T.L., A.R., J.D.M provided advice for the project. All authors reviewed, edited and approved the manuscript

## Competing interests

The authors declare no competing interests.
