## [Peer Review File · Nature Communications]

Open source microscope add-on for structured illumination microscopyREVIEWER COMMENTS

Reviewer #1 (Remarks to the Author):

The manuscript by Hannebelle et al introduces "openSIM" -- an open-source microscope add-on that allows researchers to upgrade their widefield microscopes to structured illumination super-resolution microscopes (SIM). The openSIM add-on is designed to be easily duplicated by other laboratories, making super-resolution microscopy accessible to a wider research community. Though similar open source work and SIM protocols have been demonstrated before (<https://doi.org/10.1038/ncomms10980>; <https://doi.org/10.1038/s41467-019-12165-x>; <https://www.jove.com/video/53988>), this work has additional value because it provides a feasible solution for pre-existing lab equipment without the need to build a completely new system. The add-on does not interfere with the function of other instrumentation connected to the microscope, such as incubation chambers, microscope stages, or microfluidic systems. The authors also provide documentation on their wiki-page, which I appreciate and should be encouraged. Nevertheless, I suggest key revisions before publication.

Major points:

In lines 106-117, the authors claimed that "Our goal in designing the openSIM add-on was not to compete with the most advanced SIM systems that ..." My main concern related to this statement is that a lot of technical details are not discussed in this manuscript, especially compared to conventional SIM systems. For example, the authors used high-power LEDs with much shorter coherence lengths as the excitation light sources instead of single-mode lasers, at the expense of lower modulation depth and likely coarser grating frequency applied on SLM. In general, lower modulation depth and coarser grating frequency leads to worse Wiener reconstruction (lower SNR, resolution etc.), which are important points that are not explained in the manuscript.

Several other points deserve further clarification: 1. There are no pinholes to block unwanted diffraction orders from the SLM. Are pinholes unnecessary in this setup, or are there no other diffraction orders after the SLM? 2. There is no polarization rotator to change the polarization states of the diffracted beams after the PBS to maintain s-polarized illumination at the sample for all 4 different angles. What effect is this likely to have on the quality of reconstruction?

I strongly suggest the authors clarify the consequences of the key differences between openSIM and "state-of-the-art SIM systems", perhaps in the discussion. Besides reducing costs, what are the beneficial or adverse effects of these modifications and simplifications? The answers to these questions will be important for readers to decide if they choose to clone openSIM, buy a commercial SIM or build a more advanced SIM system.

Minor points:

1. I could not find any information about the objective(s) in the main text or methods section. The only thing I know about the microscopes are they are "IX71 and IX81, Olympus". Please provide more detailed information on objective type, magnification, numerical aperture (NA) and immersion medium.
2. Camera information is also missing. Brand, model, pixel size of the images?
3. Line 61 and Supplementary Figure 4: It is unusual that in openSIM, "4 angles are used and 14 different patterns". It is well known that in linear 2D SIM, 3 angles in increments of 60° can fill the lateral Fourier plane. Each direction only needs 3 phases (i.e., pattern shifts) to decode the 0th and ±1st orders in frequency domain. That is 3 angles × 3 phases = 9 raw images in total. But openSIM uses 14 images, which would appear to increase overall exposure time and introduce unnecessary photobleaching. The authors need to explain why they choose 4 angles in increments of 45° with 3 patterns for 0° and 90° and 4 patterns for 45° and 135°.
4. Line 76: All full width at high maximum (FWHM) statistics for the PSF should also provide standard deviations (std / sd) and N.
5. Line 111-113, the authors claim that "LEDs sources, in combination with a spatial light modulator allow for a much simpler optical design, therefore the openSIM relies on incoherent LED light sources, and the pattern is generated by projection rather than interference." I do not understand why the pattern is generated by "projection" rather than interference. As openSIM still uses a ferroelectric phase-only SLM instead of a DMD to generate grating-like patterns at the imaging plane, the word "projection" does not make sense to me. My understanding is that

diffraction and interference are basic characteristics of light waves. Different light sources have different coherence lengths, which in turn affects the modulation depth at the imaging plane. It is not that SLM produces such fringes due to "projection" rather than "interference". I would be happy if the author could convince me with more physics knowledge, but more detailed and accurate explanation should be provided for this statement in the revision, or simply use 'interference'.

6. As a sanity check, please provide at least one Fourier transform amplitude image and indicate the 169 nm^{-1} position on it for Wiener reconstruction results shown in Figures or Supplementary Figures. The corresponding Wiener constant value used should also be indicated.

7. Does openSIM support time-lapse SIM imaging? If so, please provide at least one movie for live-cell imaging.

Reviewer #2 (Remarks to the Author):

Title: Review of Manuscript "OpenSIM: open source microscope add-on for structured illumination microscopy"

I was delighted to review the manuscript titled "OpenSIM: open source microscope add-on for structured illumination microscopy." Having encountered it as a preprint, I was particularly inspired by its potential and considered the possibility of implementing it myself. In our laboratory, we extensively utilize various structured illumination microscopy (SIM) setups that predominantly rely on coherent grid pattern generation using digital micromirror devices (DMDs). Consequently, I was intrigued by the concept of a setup exclusively employing incoherent illumination as it seems to remove the complexity of adjusting laser beams very exactly.

The manuscript effectively presents all the essential information, emphasizing the core concepts. The results, notably the image data, convincingly demonstrate the method's efficacy in reducing background signals and potentially improve the resolution. The associated documentation, which not only encourages replication but also thoroughly elucidates each step, establishes a benchmark for open-source hardware in scientific publications. The inclusion of video documentation is particularly commendable, and the level of detail in the online documentation is highly impressive.

A crucial point highlighted in the manuscript is the lack of integrative open-source systems that enable users to enhance commercially available microscopes, a pertinent issue often leading to redundant device development among different labs. The manuscript could potentially promote this aspect further, emphasizing the potential benefits of standardized interfaces or shared blueprints from manufacturers to foster collaborative adaptations across labs.

The authors mention that "higher resolution improvement, due to a higher achievable pattern contrast and overall illumination intensity" is achieved with 2-beam SIM. It would be valuable if the authors could delve into a comparison of the complexity and alignment effort required for constructing an incoherent SIM setup compared to a 2-beam configuration. This analysis appears to be missing from the manuscript. Also a few references and perhaps a comment on the theory behind incoherent SIM would complete the manuscript a bit.

Throughout the manuscript, several questions arose that I would like to address to the authors.

Regarding the statement, "As an open source microscope, we provide detailed documentation and instructions enabling other scientists to build their own openSIM add-on," I am curious about the target audience ("other scientists") of the manuscript. Ultimately, the audience seems to be a multidisciplinary group with a specific interest in biology, possessing a high level of technical acumen and a willingness to craft customized solutions. Although the project has achieved a high level of professionalism for a scientific "product", it remains relatively complex to construct at various points. For our laboratory, this manuscript would serve as an invaluable source of inspiration for building similar instruments, with the necessary information embedded within the instructions. It might be beneficial if the authors could mention or even elaborate on alternative components that could be used to realize the device as the ones mentioned in the text may not be

available or too expensive.

This leads to my most significant question: "these components are readily available at moderate cost from electronics supply companies and have shown good performance for SIM imaging." The manuscript adeptly demonstrates the construction of a customized video projector, combined with LabVIEW programming, to generate SIM images. Have the authors considered the possibility of directly using an off-the-shelf video projector? This approach might potentially save a substantial amount of effort, time and money. Triggering patterns on the fourth-dimension display is likely much faster than the DMD-based setup shown in a related publication. A comprehensive exploration of this topic would be appreciated. Utilizing an off-the-shelf device could greatly enhance the chances of broader adoption and community engagement.

The mention of "and linked to an open-source community" raises questions about the community's current or potential existence. The Notion document contains a wealth of information, though I experienced some difficulty navigating it due to the absence of a clear hierarchy/visible structure when departing from the main page (Disclaimer: I rarely use Notion). From my experience, it's highly advantageous when individuals can ask questions directly where problems arise. I would anticipate this to be on the GitHub page where production files are located. As of now, the design files appear to be missing, preventing any potential modifications. This, coupled with the closed documentation (precluding pull requests for improving instructions) and the use of LabVIEW as software, somewhat compromises the project's status as fully open-source—a somewhat questionable aspect. Unfortunately, I couldn't execute or test the software for potential issues due to a lack of access to LabVIEW. However, I'm sure developers have put a great effort into making the software robust and useful for biologists. This is what the photos illustrate.

With a stronger emphasis on open-source principles and community creation, an alternative software approach might have been considered. For instance, the authors could have explored options like Micromanager, ImSwitch, or Python-Microscope, many of which support most of the devices used in the setup. Such a shift might encourage wider utilization of the system.

There appears to be an omission in Line 185, concerning the mention of laser-cut parts.

Regarding the GitHub Repository:

- The mixture of formats (Notion, PDF, Markdown) for different parts of the documentation can be a bit confusing. It would be helpful to maintain a consistent format for better clarity.
- The chosen license is appropriate.
- Considering using Markdown formatting for documents could enhance readability and consistency.
- Testing/Debugging LabVIEW without significant investment in packages like the "vision" is challenging/impossible
- The file "openSIM_labview_withoutSpecVcamera_V2.0" appears to be empty
- The user manual ("userManual_openSIM_software_V2.0.pdf") is well-written and exemplary in its step-by-step approach. However, considering using Markdown instead of PDF could improve accessibility.
- Additionally in the same file:
 - o Clarify how the objective is mounted to DAQ within the user manual.
 - o Specify the data format exported, such as OME-TIFF, and provide guidance on compatibility with other software. Offering an example dataset would be valuable.
 - o Exploring the possibility of exporting the LabVIEW software as an executable could enhance usability.
 - o Clarify the purpose of indicator "I4" and its relationship with the SLM triggering the camera.
 - o The user interface is well-designed for biologists and allows for easy control.
 - o Address the unclear functionality of "Enabling/Disabling test mode." If possible, allow running the software fully without hardware or rename it to "LED Testing mode."
 - o Consider populating the troubleshooting section, offering solutions to potential issues users might face, such as wrong hardware ports or error messages.
 - o Clarify why certain options are grayed out in the settings, such as EM gain, AOI width, AOI height, and AUX output source.

- o Provide clarity on which cameras may support LabVIEW integration among the potential add-ons.
- o Consider adding a timelapse mode to the GUI.

Regarding the Electronics:

- A concise readme for the electronics section would aid in understanding the files.
- Ensure that source files are included in the electronics section.
- Provide clarification on how the LED is switched and amplified in the electronics design.
- Consider using thicker wire lines on the PCB for larger signals like those for the fan and LED.
- Adding LED spectra to match fluorophores would be beneficial, either here or in the manuscript.

Regarding the Notion Website:

- In the image gallery, clarify whether the 4 angles and 14 patterns shown represent a single reconstruction or serve as examples.
- Provide the option for raw data alongside images.
- Address the issue of images appearing small in the Notion website; consider incorporating scaling options for better visualization.

Regarding the Videos for Rebuilding the Structure:

- The videos are well-executed and comprehensive, making it feasible to recreate the setup from my technical point of view
- Question the use of a potentially expensive DIY projector; suggest considering customization of a DMD projector along with LED modifications for cost-effectiveness.

In conclusion, I find the manuscript titled "OpenSIM: open source microscope add-on for structured illumination microscopy" to be a valuable contribution to the field of microscopy, particularly in its pursuit of open-source solutions. The work showcases significant progress in reducing background signals through structured illumination microscopy and provides comprehensive documentation that encourages replication and understanding.

However, there are some critical aspects that merit attention before publication:

1. Openness and Source Availability: Given the emphasis on open-source principles throughout the manuscript, it's imperative that the associated sources and design files are equally open and accessible. Addressing the missing design files is crucial to upholding the integrity of the open-source idea. Without these files, replicating and building upon the project becomes challenging if minor modifications have to be made in order to adjust to different hardware for example. Exemplary raw data would also be helpful.

2. Documentation Format: The combination of various formats, including Notion, PDF, and Markdown, within the GitHub repository can lead to confusion for users seeking clear guidance. Standardizing the format and considering using Markdown for better readability could significantly improve the user experience. I suggest hosting most of the technical documentation on Github so that users can also interact through discussions or issues.

3. LabVIEW Dependence and Community Engagement: While the LabVIEW-based approach is valid, it does limit accessibility for those without access to LabVIEW licenses. To align with the open-source philosophy and foster broader community engagement, exploring software alternatives like Micromanager, ImSwitch, or Python-Microscope, which support a wider range of hardware, could be advantageous. This is a huge thing to ask for and I understand if this is not possible, but perhaps authors find a minimal working prototype that is e.g. provided by micromanager.

Considering these points, I recommend the manuscript for publication in principle, with the

understanding that these key issues need to be addressed to ensure the manuscript's alignment with the open-source philosophy and to enhance its reproducibility and accessibility. By actively addressing these concerns, the authors have the potential to contribute even more significantly to the open-source microscopy community and advance the field of structured illumination microscopy.

Reviewer #3 (Remarks to the Author):

I have reviewed the manuscript entitled "OpenSIM: open source microscope add-on for structured illumination microscopy" by Hannebelle et al. The manuscript describes a system for achieved structured illumination microscopy based on low cost components (3D printed parts, electronics, optics) accompanied by detailed instructions in a separated website that should enable interested readers to replicate such aa microscope. The novelty of this manuscript is moderate. It mainly depends on the fact that a very cost efficient, 3D-printable version of a SIM microscope is being demonstrated. The main paper itself only provides a general description of the system and demonstrates this by a series of examples across several different length scales. More details are found in the supplementary information file, as well as on the dedicated web site. While I am typically all in favor of publishing the information for such open systems in a high impact journal, I still have a few concerns about this manuscript in particular:

1. The name "OpenSIM" is not particularly unique. A quick google search will show that it is actually being used in a multitude of different contexts. While this by itself is not necessarily a problem, it is, however also already being used for a Matlab based SIM reconstruction algorithm described by Lal, Shan and Xi (IEEE J. Quantum Electron. 22, 50–63 (2016)), who have also just published a 3D extension of this in Nature Methods in July 2023. So, I suggest the authors find a more suitable acronym or maybe leave out the acronym entirely from the title.

2. The system requires two fairly expensive software packages, which have not been counted into the bill of materials: Matlab and LabView. Both are currently being distributed as subscription based software packages, which adds up considerably in the long-term. Thus, it would have been preferred if the authors had used entirely free and openly accessible software, e.g. written in Python.

Reviewer #1 – major points:

1.1) In lines 106-117, the authors claimed that “Our goal in designing the openSIM add-on was not to compete with the most advanced SIM systems that ...” My main concern related to this statement is that a lot of technical details are not discussed in this manuscript, especially compared to conventional SIM systems. For example, the authors used high-power LEDs with much shorter coherence lengths as the excitation light sources instead of single-mode lasers, at the expense of lower modulation depth and likely coarser grating frequency applied on SLM. In general, lower modulation depth and coarser grating frequency leads to worse Wiener reconstruction (lower SNR, resolution etc.), which are important points that are not explained in the manuscript.

Several other points deserve further clarification: 1. There are no pinholes to block unwanted diffraction orders from the SLM. Are pinholes unnecessary in this setup, or are there no other diffraction orders after the SLM? 2. There is no polarization rotator to change the polarization states of the diffracted beams after the PBS to maintain s-polarized illumination at the sample for all 4 different angles. What effect is this likely to have on the quality of reconstruction? I strongly suggest the authors clarify the consequences of the key differences between openSIM and “state-of-the-art SIM systems”, perhaps in the discussion. Besides reducing costs, what are the beneficial or adverse effects of these modifications and simplifications? The answers to these questions will be important for readers to decide if they choose to clone openSIM, buy a commercial SIM or build a more advanced SIM system.

We thank the reviewer for raising these questions and giving us an opportunity to highlight better the differences between our OpenSIM and more traditional SIM systems. These are the key points we understood from this reviewer’s comment: 1) technical/optical aspects are not discussed sufficiently / need further clarification, 2) discussion of difference between openSIM and ‘state-of-the-art’ SIM and 3) consequences of design choices.

We addressed these comments individually in the following:

1. Difference of working principle/optical design of openSIM compared to conventional SIM systems

The main difference between the optical design of openSIM and other conventional systems is the excitation pattern generation approach and the use of an incoherent light source. Conventional SIM systems use high-frequency fringe patterns formed by interference of diffraction beams in the sample plane to illuminate the sample. For this, commonly LCOS SLMs in phase-only mode are used. Coherent light is collimated onto the SLM, which serves as a phase grating, leading to the diffraction beams. The illumination pattern is then created by the interference of two diffraction beams (for 2D SIM) or three diffraction beams (for 3D SIM) in the sample plane. The approach for the illumination pattern formation of openSIM follows a non-diffraction or interference-based method using non-coherent light, described in¹⁻⁵, in which an image of the SLM is formed directly in the sample plane instead of being created by interference of diffraction beams. For this, the LCOS SLM is placed in the primary image plane of the microscope and used in combination with a polarizing beam splitter (PBS) in amplitude mode to serve as a computer-controlled binary mask. The microdisplay is imaged onto the sample via a polarizing beam splitter, an additional tube lens and the microscope’s objective.

→ We included this information in a concise format in the interest of readability, highlighting the key details within the manuscript's discussion and methods sections. Additionally, we presented a comprehensive version of this information, along with additional schematics, in the documentation

(corresponding page of openSIM website, section 'LCOS in amplitude mode', 'Difference in the optical design of openSIM compared to conventional SIM systems').

2. Advantages / disadvantages of used optical design compared to conventional SIM systems

One of the primary benefits of employing direct imaging of the pattern into the sample plane, as opposed to systems relying on interference, lies in the inherent simplicity of the optical setup with a small number of basic optical components and reduced demands for precise positioning and alignment of these optical elements, making the overall system more robust and less prone to alignment-related challenges.

An additional advantageous facet of this setup lies in the flexibility regarding the usable wavelengths and desired patterns, enabling the switching between multiple wavelengths and the use of different pattern orientation angles. The SLM possesses the capability to create arbitrary binary patterns, and notably, the spatial frequency of these patterns remains independent of the wavelength in use³. Moreover, the optical design does not necessitate components with a wavelength- or pattern orientation-dependent layout (e.g. phase-plane masks, rotating polarizer). As a consequence, such components demand for either a setup with preset configurations (i.e. accommodation of only one fixed wavelength and set of pattern orientations) or more complex optical components.

A primary objective in SIM imaging is to achieve the highest possible contrast (degree of modulation) in the excitation pattern at the sample plane, aiming for an optimal signal-to-noise ratio⁶. Despite the described advantages of the chosen optical design of openSIM, it presents certain drawbacks that potentially result in a reduced image contrast.

Placing the microdisplay in the image plane for direct pattern imaging into the sample plane results in an inefficient use of the illumination light. Nonetheless, the employed high-power LEDs possess sufficient brightness to offset this inefficiency¹. Furthermore, polarizing elements such as the LCOS display and the PBS cube exhibit suboptimal polarization properties leading to a decrease of the pattern contrast⁷.

→ We included this information in a concise format in the interest of readability, highlighting the key details within the manuscript's discussion and methods sections. Additionally, we presented a comprehensive version of this information in the documentation (corresponding page of openSIM website, section 'Difference in the optical design of openSIM compared to conventional SIM systems').

3. Omission of pinholes in optical design to block unwanted diffraction order from the SLM

Omission of polarization rotator in optical design to change the polarization states of the diffracted beams to maintain s-polarized illumination at the sample for all 4 different angles

For SIM systems in which the illumination pattern is generated based on interference, the appropriate handling of diffraction orders and maintenance of polarization states is crucial. To attain the highest level of modulation, it is beneficial to obstruct the zero-diffraction order coming from non-diffracted light for 2D SIM⁶ and unwanted higher diffraction orders caused by the finite-sized pixels of the SLM⁸. This can be achieved by a mask with pinholes at the appropriate position in the illumination path (i.e. pupil plane)⁸⁻¹⁰.

Furthermore, for the interference of beams, the light needs to have the same polarization¹¹. To maximize the pattern contrast, it has to be assured that the illumination light is s-polarized for all pattern orientations⁹, which can be achieved by a rotating polarizer^{8,9,11} or a patterned polarizer¹⁰. The implementation of such a pupil-plane mask and polarizer can be challenging since they have to be designed and precisely positioned according to the used wavelength and pattern orientations.

For SIM system based on direct imaging of pattern into the sample plane, pinholes to block unwanted diffraction and polarization rotators to maintain s-polarized illumination, normally necessary for pattern generation based on interference, are not required, which is reducing the complexity of the optical design.

Although a pinhole may not be considered an essential component in our case, it could potentially enhance the quality of the pattern by serving as a spatial filter and effectively blocking unwanted

higher diffraction orders or unwanted stray light from other sources. Nevertheless, the integration of a pinhole would necessitate the inclusion of supplementary optical elements, demanding precise positioning within the optical system, and would result in a more complex optical path. Since we are aiming for a simple and compact add-on, easily mountable to microscopes which often have setup-inherent space constraints, we opted not to incorporate an additional pinhole to maintain the simplicity and compactness of our design. However, undesired diffraction orders higher than the 1st order stemming from the finite size of the microdisplay's pixel are naturally blocked by the design of the 3D printed optics block.

→ We included this information in a concise format in the interest of readability, highlighting the key details within the manuscript's discussion and methods sections. Additionally, we presented a comprehensive version of this information in the documentation (corresponding page of openSIM website, section 'Difference in the optical design of openSIM compared to conventional SIM systems').

5. Use of incoherent light source (disadvantages/advantages, limitations)

A main factor influencing the resolving performance of SIM is the highest achievable illumination pattern contrast. Non-coherent systems^{1-5,12} such as openSIM, which employ incoherent light from high power LEDs, exhibiting a reduced maximal attainable contrast which consequently leads to a diminished enhancement in resolution when compared to coherent systems^{8-10,13-16} utilizing coherent laser light. While for coherent illumination SIM, the coherent optical transfer function applies, where the pattern contrast remains constant with increasing spatial frequency, for incoherent illumination SIM, the incoherent optical transfer function applies, causing pattern contrast to decrease as spatial frequency increases¹⁷, leading to a diminished contrast for high-frequency patterns.

Nevertheless, despite the potential higher achievable resolution of coherent illumination SIM, the use of coherent light sources comes with several drawbacks. Since, we were aiming to provide an inexpensive and compact system which is straightforward to reconstruct, aspects such as cost-effectiveness, availability, small dimensions and safety of the used components were of main importance. Lasers are expensive, spacious and only available for certain wavelengths, whereas LEDs are less limited in terms of available wavelengths, making them a cost-efficient compact choice, simplifying the incorporation of multiple excitation channels and the exchange of the light source in order to match the different spectra of other fluorophores. Moreover, lasers are generally less safe to use than LEDs as they pose a higher risk of eye damage due to their high intensity. Furthermore, in contrast to LEDs which exhibit a low spatial coherence, the high coherence of laser light can be problematic due to stray interference patterns. This can be circumvented by spatially scrambling the light, however, increasing complexity and costs⁶.

→ We included this information in a concise format in the interest of readability, highlighting the key details within the manuscript's discussion sections. Additionally, we presented a comprehensive version of this information in the documentation (corresponding page of openSIM website, section 'Difference in the optical design of openSIM compared to conventional SIM systems').

Reviewer #1 – minor points:

1.2) I could not find any information about the objective(s) in the main text or methods section. The only thing I know about the microscopes are they are "IX71 and IX81, Olympus". Please provide more detailed information on objective type, magnification, numerical aperture (NA) and immersion medium.

Camera information is also missing. Brand, model, pixel size of the images?

We apologize for this oversight on our part. An 'image acquisition' paragraph was added to the methods section in the manuscript containing information about the used objectives and cameras during image acquisition.

Manuscript methods section 'Image acquisition': The fixed mouse intestinal organoid, the live zebrafish embryo, the macrophages infected with Mycobacterium smegmatis bacteria and the fixed actin-stained COS-7 cells were imaged using a 20x and 100x (1.3 NA, oil immersion) objective. The 100nm bead sample and the fixed bovine pulmonary artery endothelial cells were imaged with a 100x objective (1.3 NA, oil immersion). For the image acquisition an Andor Zyla 4.2 Plus camera (pixel format 1048x1048) and an Andor iXon3 camera (pixel format 1024x1024) were used. We provide example data sets of the fixed 100nm bead sample, fixed bovine pulmonary artery endothelial cell sample and the live zebrafish sample on Zenodo including the raw data and the reconstructed data with corresponding information about the used parameter setting during image acquisition and image reconstruction.

1.3) Line 61 and Supplementary Figure 4: It is unusual that in openSIM, "4 angles are used and 14 different patterns". It is well known that in linear 2D SIM, 3 angles in increments of 60° can fill the lateral Fourier plane. Each direction only needs 3 phases (i.e., pattern shifts) to decode the 0th and ±1st orders in frequency domain. That is 3 angles × 3 phases = 9 raw images in total. But openSIM uses 14 images, which would appear to increase overall exposure time and introduce unnecessary photobleaching. The authors need to explain why they choose 4 angles in increments of 45° with 3 patterns for 0° and 90° and 4 patterns for 45° and 135°.

An 'illumination pattern' section was added to the methods section in the manuscript giving general information about SIM illumination pattern, pattern design considerations and explanation about the chosen pattern sequences.

Manuscript methods section 'Illumination pattern': In order to obtain nearly isotropic resolution improvement in 2D linear SIM, the periodic pattern needs to be shifted through three phases (120° to each other) and for each of three orientations (0°, 60°, 120°), yielding nine raw images¹³. In our experiments, a sequence of periodic line patterns with 3 orientation angles and 4 images per angle (12 different patterns in total) was used, in total resulting in a homogenous illumination. The set of patterns was designed according to Lukeš et al.² so that the pattern for different orientation angles have a similar mark-to-area ratio (MAR) (i.e. fraction of the illumination area)⁶, and spatial frequency. Because the pixels on the microdisplay have a square shape, additional patterns were used in the diagonal directions to ensure even coverage across the entire image while maintaining a roughly consistent spatial frequency for the patterns². While three angles are sufficient for SIM image reconstruction, using more angles can potentially increase the spatial resolution and improve the robustness and accuracy of the image reconstruction. Nevertheless, this comes at the expense of extended acquisition times, increased photobleaching and computational cost. We thus used a pattern repertoire with an additional angle orientation (4 angles and 3 or 4 images per angle, 14 different patterns in total) for non-biological or fixed samples where photobleaching is not of importance.

1.4) Line 76: All full width at high maximum (FWHM) statistics for the PSF should also provide standard deviations (std / sd) and N.

Standard deviation and N were added to all FWHM value in figure 2 and supplementary figure 8.

1.5) Line 111-113, the authors claim that "LEDs sources, in combination with a spatial light modulator allow for a much simpler optical design, therefore the openSIM relies on incoherent LED light sources, and the pattern is generated by projection rather than interference." I do not understand why the pattern is generated by "projection" rather than interference. As openSIM still uses a ferroelectric phase-only SLM instead of a DMD to generate grating-like patterns at the imaging plane, the word "projection" does not make sense to me. My understanding is that diffraction and interference are basic characteristics of light waves. Different light sources have different coherence lengths, which in turn affects the modulation depth at the imaging plane. It is not that SLM produces such fringes due to "projection" rather than "interference". I would be happy if the author could convince me with more

physics knowledge, but more detailed and accurate explanation should be provided for this statement in the revision, or simply use 'interference'.

openSIM follows a different approach compared to conventional SIM systems which indeed create the excitation pattern by interference of diffraction beams often by utilizing a LCOS SLM. openSIM also uses a LCOS SLM to generate the illumination pattern but in amplitude mode instead of phase-only mode. In contrast to interference-based systems, openSIM generates the pattern by directly forming an image of the microdisplay, which is placed in the primary image plane of the microscope, in the sample plane.

In this context (formation of an image of the microdisplay in sample plane) the usage of the term projection is adequate in our opinion.

But indeed, the different working principle of openSIM compared to conventional SIM was not explained sufficiently. To avoid any ambiguity and to emphasize the differences in the working principle and used components compared to conventional systems, more information about the pattern generation were added to the manuscript discussion and methods section (see above) and a detailed explanation with schematics was added to the documentation (corresponding page of openSIM website, section 'LCOS in amplitude mode', 'Difference in the optical design of openSIM compared to conventional SIM systems')

1.6) As a sanity check, please provide at least one Fourier transform amplitude image and indicate the 169 nm⁻¹ position on it for Wiener reconstruction results shown in Figures or Supplementary Figures. The corresponding Wiener constant value used should also be indicated.

A figure (supplementary figure 12) with the Fourier transforms of an SIM image and corresponding WF image with circular lines indicating the approximately achieved resolution was created and added to the manuscript. The corresponding Wiener constant used for image reconstruction was indicated.

The used Wiener constant was indicated in the image reconstruction parameter read-me file for all the exemplary datasets.

1.7) Does openSIM support time-lapse SIM imaging? If so, please provide at least one movie for live-cell imaging.

The openSIM software did not support time-lapse imaging so far, but we implemented a time-lapse mode within the scope of the revision process (see supplementary figure 5). We have added a section to the user manual explaining the timelapse functionality of the software (see supplementary figure 5). We have not acquired time-lapse images so far.

Reviewer #2 – major points

2.1) Openness and Source Availability: Given the emphasis on open-source principles throughout the manuscript, it's imperative that the associated sources and design files are equally open and accessible. Addressing the missing design files is crucial to upholding the integrity of the open-source idea. Without these files, replicating and building upon the project becomes challenging if minor modifications have to be made in order to adjust to different hardware for example. Exemplary raw data would also be helpful.

We apologize that the setting of the openSIM GitHub repository, containing the design files, was accidentally not yet set to public in the beginning of the review process leading to access problems and that there was a file missing. We now assured that all design files necessary for building up openSIM are uploaded and publicly accessible.

We uploaded exemplary raw datasets of different sample types (100nm beads, fixed tubulin, zebrafish z-stack) onto Zenodo containing the raw data and reconstructed data with corresponding image acquisition and image reconstruction parameter.

2.2) Documentation Format: The combination of various formats, including Notion, PDF, and Markdown, within the GitHub repository can lead to confusion for users seeking clear guidance. Standardizing the format and considering using Markdown for better readability could significantly improve the user experience. I suggest hosting most of the technical documentation on Github so that users can also interact through discussions or issues.

We thank the reviewer for these suggestions. We partly agree with the reviewer's comments that the use of various formats (PDF, pages on Notion, read-me files on GitHub) and the use of different platforms (Notion, GitHub) can potentially lead to unclarity about the documentation structure and where to find specific content. However, we indeed considered several formats and platforms (lab webpage, GitHub repository with read-me files and internal wiki, Github webpage, Notion) to host our documentation and weighted the advantages and disadvantages before deciding for one. We consciously chose each option due to its benefits in the specific application in contrast to the other options. In the end we decided to use a website (Notion) for information about the assembly, operation and background theory (all information) and a repository (GitHub) where all technical design files are provided (all files). The primary reason for using Notion is the simple handling without the need for any coding experience, the straightforward option to publicly share the content as a webpage and to add editors to the Notion project enabling other user to modify the content of the documentation. We considered to host the documentation on a Github webpage which enables users to directly modify the documentation or to fork the underlying repository. However, in order to do the user requires additional knowledge about the structure of the repository and basic knowledge about the used syntax making any modifications more complicated. A GitHub repository was chosen to upload all the technical design files since it allows the user to easily download the files individually or to fork the whole repository. Despite the lack of version control for CAD files or LabVIEW, GitHub offers better options to track the versions of the uploaded files.

In response to the reviewer's comments, we improved the clarity and navigation of the documentation in the following way: In order to present the structure of the documentation and to guide the user to the different parts of the documentation we added more information about the structure and content with a corresponding schematic to the openSIM webpage (corresponding page of openSIM website) and additional read-me files explaining the structure of the openSIM repository to the GitHub repository. Furthermore, we added more information to the read-me files on GitHub.

On the openSIM Notion webpage, we added more dividers between the different paragraphs and included more toggle structures to differentiate the different topics on each Notion pages to improve the readability (exemplary openSIM website page with improved structure).

2.3) LabVIEW Dependence and Community Engagement: While the LabVIEW-based approach is valid, it does limit accessibility for those without access to LabVIEW licenses. To align with the open-source philosophy and foster broader community engagement, exploring software alternatives like Micromanager, ImSwitch, or Python-Microscope, which support a wider range of hardware, could be advantageous. This is a huge thing to ask for and I understand if this is not possible, but perhaps authors find a minimal working prototype that is e.g. provided by micromanager.

We thank the reviewer for this comment and agree with the reviewer that the required LabVIEW licenses are a significant hurdle for people who do not have access to site licenses. To share an open-source project with a wide community while providing users the freedom to debug and modify the software, the most suitable choice is to distribute a software developed using an open-source programming language. However, this requires that the user has experience with coding and knows the used programming language. Furthermore, the user needs to understand the structure of the

software, which is more complicated when the software contains a GUI, event-based structure and communication with external hardware devices.

Our goal with our open hardware projects is to not only make it easy for people to copy and use, but also to modify and expand on. Making the software easy to understand and modify is a big part of that. The “beauty” of LabVIEW is that it enables seamless connection of the software to data acquisition hardware, which is much more challenging for novices in Python or C. In our experience, it is much easier for non-instrumentation-enthusiasts to learn how to make small modifications in LabVIEW than in other languages. We early on considered writing the software in Python, and also looked into micromanager. However, we found that this would make the software much more difficult to modify for novices. National instruments now offers also a LabVIEW Community Edition which is free for non-commercial, non-academic use and has all the capabilities of the LabVIEW Professional edition.

To nevertheless make it possible to for interested readers to build and operate the OpenSIM, even if they not have access to a full LabVIEW license, we now offer three software versions to cater to various user requirements (see corresponding page of openSIM website section ‘openSIM software’ for more information):

- **Full LabView Project Versions:** For users seeking to modify or customize the software, we provide two full LabView projects. The first version, "openSIM_software_withCameraControl," integrates complete camera control, including image acquisition and data handling, specifically designed for Andor Zyla and Andor iXon camera models.
- The second version maintains the same structure but omits camera-model-specific components, **making it modular and adaptable for other camera models.**
- **openSIM Application (exe):** To overcome the need for any chargeable licenses and to offer a free software option, we created an openSIM software version which allowed us to build an application from it. This software version is designed for user who primarily want to use the software without any need for modification. Running the software version does not require any paid software licenses and is not tied to a specific camera model.

We functionally tested this application on an PC without any pre-installed LabVIEW modules to ensure a faultless function despite having not any LabVIEW license. We also created an additional user manual (corresponding page on openSIM website) particular to this software version with specific installation guide and troubleshooting section.

In the future we are envisioning to offer an openSIM software in pure Python environment. For this we are planning to use the Python package pymmcore_plus (Python binding for Micromanager with C++ core) to control the camera via a camera specific Micromanager device adapter. The packet pymmcore-widgets offers a set of widgets for the pymmcore-plus package to build a custom user interface. Additional libraires can be used to control the other external devices (e.g. NI-DAQ Python API). Qt designer can be used to develop the event structure. Our first aim would be to provide a python library which offers all the functionalities of the current software. In a next step we would build a full GUI which improves the user experiences but also requires additional effort. Nevertheless, as the reviewer already pointed out, developing the openSIM software entirely within a Python environment from scratch requires a significant amount of work. Thus, we cannot provide this version in the scope of this revision process.

Nonetheless, we have attempted to address this by offering an executable version of the openSIM software.

Reviewer #2 – minor points

We want to thank reviewer #2 for the great amount of helpful and especially detailed feedback about the openSIM website and user manual. Addressing all the comments significantly improved the quality of the documentation potentially helping to motivate others to build up their own openSIM.

2.4) The manuscript could potentially promote this aspect further, emphasizing the potential benefits of standardized interfaces or shared blueprints from manufacturers to foster collaborative adaptations across labs.

In response to the reviewer's comment, we further elaborated this aspect in the discussion of the manuscript.

Manuscript discussion section: We aimed for an add-on which facilitates an effortless integration with various commercially available microscopes (such as Olympus, Zeiss, Leica) through the straightforward replacement of the tube lens and illumination port adapter. However, it's worth considering that the microscope-specific components may limit versatility. To potentially enhance versatility, manufacturers might consider promoting standardized interfaces and sharing blueprints to possibly reduce the necessity for custom adapters or connectors. This could offer laboratories a valuable resource for constructing and tailoring hardware components which could simplify the setup and streamline workflows for researchers. Consequently, this might encourage more laboratories to openly share their work, foster collaborative adaptations, and lessen the barriers that could impede other labs from adopting open-source devices developed by different labs.

2.5) It would be valuable if the authors could delve into a comparison of the complexity and alignment effort required for constructing an incoherent SIM setup compared to a 2-beam configuration. This analysis appears to be missing from the manuscript. Also a few references and perhaps a comment on the theory behind incoherent SIM would complete the manuscript a bit.

This aspect was also raised by reviewer #1. We thank the reviewer for this comment and agree that the differences in the optical design, especially in respect to the underlying theory, complexity and alignment, was not discussed sufficiently.

In order to address this comment, we added a comparison of complexity and alignment effort between openSIM and conventional SIM systems in a concise format in the interest of readability, highlighting the key details within the manuscript's discussion and methods sections. Additionally, we presented a comprehensive version of this information in the documentation (corresponding page on openSIM website, sections: 'Difference in the optical design of openSIM compared to conventional SIM systems', 'Differences in the optical alignment compared to conventional SIM systems'). In this context we also elaborate on the theory behind incoherent SIM systems based on interference of diffraction beams. We also delve into the advantages and disadvantages of the simple optical design of openSIM compared to conventional SIM systems both in the manuscript and in more detail in the documentation.

Please find more information in the response discussed above (1.1).

2.6) It might be beneficial if the authors could mention or even elaborate on alternative components that could be used to realize the device as the ones mentioned in the text may not be available or too expensive.

We added an additional section in the documentation (corresponding page of openSIM website, section: 'Note: Alternative products') where we introduce and evaluate different alternative options to the used components, in case the user envisions a cheaper option or the specific components is not offered by the manufacturer anymore. However, we cannot guarantee that these alternative components will work as plug-in replacements.

2.7) With a stronger emphasis on open-source principles and community creation, an alternative software approach might have been considered. For instance, the authors could have explored options

like Micromanager, ImSwitch, or Python-Microscope, many of which support most of the devices used in the setup. Such a shift might encourage wider utilization of the system.

Please refer to the response discussed above (2.3).

2.8) There appears to be an omission in Line 185, concerning the mention of laser-cut parts.

Information about the laser-cut parts was added to the methods section of the manuscript.

We used a 3mm thick black acrylic sheet for the panels of the enclosure. The sheets were trimmed to the corresponding size with a laser cutter.

2.9) The mixture of formats (Notion, PDF, Markdown) for different parts of the documentation can be a bit confusing. It would be helpful to maintain a consistent format for better clarity.

Considering using Markdown formatting for documents could enhance readability and consistency.

We acknowledge that the combination of various formats can create potential confusion. Unfortunately, for the Notion website, we are restricted to using specific pre-defined formatting options. While Notion offers a user-friendly and straightforward approach, it comes with the drawback of limited customization choices. Having a PDF version available greatly helps when assembling the instrument in the workshop. We formatted the PDF with the intention of enhancing the document's structure, clarity and readability, even though the formatting may deviate from the guidelines outlined in the GitHub repository's readme files. We ensured that all readme files on GitHub have the same markdown formatting.

2.10) Testing/Debugging LabVIEW without significant investment in packages like the "vision" is challenging/impossible

We thank the reviewer for this comment. Indeed, testing and debugging the full LabVIEW projects is not possible without the LabVIEW vision development module. However, the openSIM application can be tested without any chargeable license. We added a table listing all the functionalities and debugging/modification options for the individual software versions to the documentation (see corresponding page of openSIM website, section 'openSIM software').

2.11) The file "openSIM_labview_withoutSpecVlcamera_V2.0" appears to be empty

We apologize for the missing file. We now assured that all design files necessary for building up openSIM are uploaded and publicly accessible in the GitHub repository.

2.12) The user manual ("userManual_openSIM_software_V2.0.pdf") is well-written and exemplary in its step-by-step approach. However, considering using Markdown instead of PDF could improve accessibility.

We thank the reviewer for the positive feedback of our user manual and for the suggestion. We specifically decided to provide the user manual in pdf format since we thought this could simplify using it during an actual imaging session as the pdf can be printed and read parallel in paper format while the openSIM software user interface is open on the screen.

2.13) Clarify how the objective is mounted to DAQ within the user manual

More general information about the z-actuator and additional explanation on how to mount the objective to the z-actuator with schematics and annotated images was added to the corresponding section of the user manual (user manual, section 'Appendix – Peripheral components (z-actuation)')

2.14) Specify the data format exported, such as OME-TIFF, and provide guidance on compatibility with other software. Offering an example dataset would be valuable.

In order to address the reviewer's comment, we added more information to the user manual about the data format and data handling with an annotated exemplary image of the data folder structure (user manual, section '7. Data handling'). We also explain data format compatibility with the

SIMToolbox reconstruction software and the required data and metadata structure to facilitate an automatic read-in by the software.

Furthermore, we included an additional section to the user manual where we explain the structure of the distributed software folder and included files (user manual, section '4. Distributed files and required folder structure)

2.15) Exploring the possibility of exporting the LabVIEW software as an executable could enhance usability.

We created an executable openSIM software version with corresponding user manual which we provide on the GitHub repository (see more information above)

2.16) Clarify the purpose of indicator "I4" and its relationship with the SLM triggering the camera.

We added more information about indicator 14 to the user manual to clarify usage (now after some changes indicator 15 in current user manual version).

2.17) Address the unclear functionality of "Enabling/Disabling test mode." If possible, allow running the software fully without hardware or rename it to "LED Testing mode."

We already implemented a software mode which enables running the software without hardware. The debugging modes is to facilitate testing and debugging of the software and the individual devices (i.e. SLM, NI DAQ, camera). It can be useful to connect only some of the devices (e.g. only NI DAQ is connected and tested with the software, while the SLM and/or camera is not connected). However, in the default (non-debugging) mode the openSIM software will return an error message and stops when devices are not connected. In the debugging mode, the software can be used despite the corresponding device not being connected.

The term 'test mode' is indeed inaccurate as it is only a fan test mode. We renamed the control in the software user interface and explained the mode clearer in the user manual.

2.18) Consider populating the troubleshooting section, offering solutions to potential issues users might face, such as wrong hardware ports or error messages.

In response to the reviewer's comment, we added further information to the existing troubleshooting cases and added additional troubleshooting cases. We also created a troubleshooting section specific for the user manual for the executable software version, since this software version could potentially encounter different errors. We also added more custom error message or user prompts to the openSIM software for cases in which the user connects / selects something wrong. This potentially prevents LabView internal error messages leading to a more uncontrolled shutdown of the software or missing functionalities.

2.19) Clarify why certain options are grayed out in the settings, such as EM gain, AOI width, AOI height, and AUX output source.

For a better user-experience and an easier handling of the openSIM software we made the visibility or color (disabled and greyed out) of certain controls and indicators depended on the availability of the corresponding device (e.g. z-stacking option is only visible/enabled if a z-actuator as peripheral component is selected or EM gain is only visible/enabled if an EMCCD camera is selected) or depended on certain selections (e.g. control and indicators corresponding to the automatic pattern selection mode are only visible/enabled if the automatic mode and not the manual mode is selected). We added more information to the user manual to clarify this aspect.

2.20) Provide clarity on which cameras may support LabVIEW integration among the potential add-ons.

We clarified the camera compatibility in the manuscript methods section, the user manual (user manual, section '1. Introduction to the openSIM software') and the documentation (corresponding page of the openSIM website, section 'openSIM software'). We also pointed out the camera-model independent implementation of the openSIM software executable.

Manuscript methods section 'LabVIEW interface and openSIM control): At the moment we offer several software versions with different functionalities in order to provide an adequate option for user with diverse requirements. For users who want to modify or customize the software we provide the full LabVIEW project including camera control (exposure time, camera gains, binning options) and data handling (pre-formatting the data to be compatible directly with the open source SIMToolbox SIM processing algorithm). The first version includes a full camera control for the camera models Andor Zyla PLUS sCMOS (6.5 μ m pixel size) and Andor iXon3 EMCCD (8 μ m pixel size). The modular design of the software provides the possibility to include the control of other commercial cameras. For user who want to primarily use the software, we offer an application (exe) of the openSIM software which is not camera model specific.

2.21) Consider adding a timelapse mode to the GUI.

We implemented a timelapse mode in the openSIM software (see supplementary figure 5) and added a corresponding section to the user manual.

2.23) A concise readme for the electronics section would aid in understanding the files.

We added a readme file to the electronics folder on the openSIM GitHub repository with explanation of the individual files and their purpose and an annotated image of the folder structure.

2.24) Ensure that source files are included in the electronics section.

We ensured that the source files are also included in the electronics section.

2.25) Provide clarification on how the LED is switched and amplified in the electronics design.

We created an additional figure (supplementary figure 10) explaining the synchronization and the timing scheme of LED, SLM and camera and explained the background why a synchronization of these components is necessary. We added more information about the LEDs and the LED driver boards to the documentation (corresponding page of openSIM website, section 'LEDs'). The actual LED amplification is done with a compatible commercial LED driver board. In the corresponding section in the documentation, we linked the LED and LED driver board specification sheet.

Manuscript methods section 'Synchronization camera, SLM and LEDs': SIM images can exhibit artifacts associated with the illumination pattern. Thus, the timing of the system's sub-components, namely camera, SLM, LED, has to be precisely synchronized (Supplementary Figure 10). The camera's exposure signal is employed to initiate the sequential projection of the patterns chosen within the pattern sequence. LCOS microdisplays cannot continuously display patterns since each pixel state must be inverted after every image to avoid a charge build-up. To facilitate this, the illumination source must be temporarily disabled during the refresh cycles, necessitating a precisely synchronized timing scheme for both the microdisplay and LEDs. The time period, which delineates the intervals for displaying the pattern or the inverted pattern, is determined by predefined timing schemes provided by the manufacturer of the SLM (ForthDimension).

Manuscript methods section 'Electronic components for the openSIM': The LEDs are powered and controlled using the boards and connectors supplied in the DK114N3 development kit (Luminus devices). The kit consists of high current driver boards and cable assemblies for each channel to provide the LED drive current for high brightness. A custom interconnect board was designed to facilitate the connection between the elements of the openSIM, such as the data acquisition device, LEDs, thermistor and fan for temperature control, SLM and camera.

2.26) Consider using thicker wire lines on the PCB for larger signals like those for the fan and LED. We thank the reviewer for the suggestion. The power supply for the fan and LED does not go over the interconnect PCB. The PCB only provides enable and PWM signals for the LED and fan. For the power supply cables with appropriate dimensions are used. Additional information and a schematic presenting the function of the interconnect board was added to the documentation to improve the understanding (corresponding page of openSIM website, section: 'Interconnect PCB')

2.27) Adding LED spectra to match fluorophores would be beneficial, either here or in the manuscript. Additional information about the used LEDs (emitting area, dominating wavelengths) and the spectral distribution was added to the manuscript methods section and the documentation (corresponding page of the openSIM webpage, section 'LEDs'). An additional figure (supplementary figure 11) was created depicting the LED spectra.

Manuscript methods section 'Optical components for the openSIM': Three high power PT54 LEDs (PT-54-TE, Luminus Devices; dominant wavelengths: red-amber $\lambda_d = 613\text{nm}$, green $\lambda_d = 525\text{nm}$, blue $\lambda_d = 460\text{nm}$, emitting area of 5.4 mm^2) are used to generate the illumination light (Supplementary Figure 11).

2.28) In the image gallery, clarify whether the 4 angles and 14 patterns shown represent a single reconstruction or serve as examples.

More information was added to the corresponding figure caption to clarify this aspect. We also included an additional section in the user manual (section '5. Illumination patterns') where we give more background information on the creation of the provided pattern sequences (repz files) and the handling and automatic saving option in the openSIM software to enable an automatic read-in of pattern related information by the SIMToolbox software necessary for SIM image reconstruction.

2.29) Provide the option for raw data alongside images.

We uploaded exemplary raw datasets of different sample types (100nm beads, fixed tubulin, zebrafish z-stack) onto Zenodo containing the raw data and reconstructed data with corresponding image acquisition and image reconstruction parameter.

2.30) Address the issue of images appearing small in the Notion website; consider incorporating scaling options for better visualization.

The scaling of images in Notion was improved and the sizes of the images were adjusted to each other. In case there was more than one image per point, the images were inserted as one image after editing to improve scaling. Furthermore, more annotations were added to the images to further improve the understanding.

2.31) Question the use of a potentially expensive DIY projector; suggest considering customization of a DMD projector along with LED modifications for cost-effectiveness.

In order to address the reviewer's comment, we discussed this aspect in the manuscript method section.

Manuscript method section 'Optical design of openSIM': LCOS are most commonly used for SLM-based pattern generation due to their high level of performance. The use of DMDs for SIM¹⁸⁻²¹ would be an alternative. They are more cost-effective and the implementation of the timing scheme is less difficult. Conversely, LCOS microdisplays exhibit reduced diffractive efficiency, resulting in fewer diffraction losses into higher orders that may not be captured by the imaging lens²².

3.1) The name "OpenSIM" is not particularly unique. A quick google search will show that it is actually being used in a multitude of different contexts. While this by itself is not necessarily a problem, it is, however also already being used for a Matlab based SIM reconstruction algorithm described by Lal, Shan and Xi (IEEE J. Quantum Electron. 22, 50–63 (2016)), who have also just published a 3D extension of this in Nature Methods in July 2023. So, I suggest the authors find a more suitable acronym or maybe leave out the acronym entirely from the title.

We thank the reviewer for the feedback. To avoid any ambiguity, the name openSIM in title was exchanged with open-source SIM, in response to the reviewer's comment.

3.2) The system requires two fairly expensive software packages, which have not been counted into the bill of materials: Matlab and LabView. Both are currently being distributed as subscription based software packages, which adds up considerably in the long-term. Thus, it would have been preferred if the authors had used entirely free and openly accessible software, e.g. written in Python.

This aspect was also raised by reviewer #2. We thank the reviewer for this insightful comment and completely agree with the reviewer's opinion. To address the reviewer's comment, we developed an additional software version based on which we build an application which we now provide on the openSIM repository

Please refer to the response discussed above for more information (2.3).

References

1. Křížek, P., Raška, I. & Hagen, G. M. Flexible structured illumination microscope with a programmable illumination array. *Opt. Express* **20**, 24585 (2012).
2. Lukeš, T. *et al.* Three-dimensional super-resolution structured illumination microscopy with maximum a posteriori probability image estimation. *Opt. Express* **22**, 29805 (2014).
3. Pospíšil, J. *et al.* Imaging tissues and cells beyond the diffraction limit with structured illumination microscopy and Bayesian image reconstruction. *GigaScience* **8**, giy126 (2019).
4. Schlichenmeyer, T. C., Wang, M., Elfer, K. N. & Brown, J. Q. Video-rate structured illumination microscopy for high-throughput imaging of large tissue areas. *Biomed. Opt. Express, BOE* **5**, 366–377 (2014).
5. Křížek, P., Lukeš, T., Ovesný, M., Fliegel, K. & Hagen, G. M. SIMToolbox: a MATLAB toolbox for structured illumination fluorescence microscopy. *Bioinformatics* **32**, 318–320 (2016).
6. Heintzmann, R. Structured illumination methods. in *Handbook of Biological Confocal Microscopy* 265–279 (Springer, 2006).
7. Křížek, P. & Hagen, G. Spatial light modulators in fluorescence microscopy. in 1366–1377 (2010).
8. Kner, P., Chhun, B. B., Griffis, E. R., Winoto, L. & Gustafsson, M. G. L. Super-resolution video microscopy of live cells by structured illumination. *Nat Methods* **6**, 339–342 (2009).
9. Shao, L., Kner, P., Rego, E. H. & Gustafsson, M. G. L. Super-resolution 3D microscopy of live whole cells using structured illumination. *Nat Methods* **8**, 1044–1046 (2011).
10. Förster, R. *et al.* Simple structured illumination microscope setup with high acquisition speed by using a spatial light modulator. *Opt Express* **22**, 20663–20677 (2014).
11. Rego, E. & Shao, L. Practical Structured Illumination Microscopy. *Methods in molecular biology (Clifton, N.J.)* **1251**, 175–92 (2015).
12. Heintzmann, R. & Cremer, C. Laterally modulated excitation microscopy: Improvement of resolution by using a diffraction grating. *Proc. SPIE* **3568**, (1999).

13. Gustafsson, M. G. L. Surpassing the lateral resolution limit by a factor of two using structured illumination microscopy. *J Microsc* **198**, 82–87 (2000).
14. Gustafsson, M. G. L. *et al.* Three-Dimensional Resolution Doubling in Wide-Field Fluorescence Microscopy by Structured Illumination. *Biophysical Journal* **94**, 4957–4970 (2008).
15. Fiolka, R., Shao, L., Rego, E. H., Davidson, M. W. & Gustafsson, M. G. L. Time-lapse two-color 3D imaging of live cells with doubled resolution using structured illumination. *Proceedings of the National Academy of Sciences* **109**, 5311–5315 (2012).
16. Lu-Walther, H.-W. *et al.* fastSIM: a practical implementation of fast structured illumination microscopy. *Methods Appl. Fluoresc.* **3**, 014001 (2015).
17. Goodman, J. W. Frequency Analysis of Optical Imaging Systems. in *Introduction to Fourier Optics* (1996).
18. Sandmeyer, A. *et al.* Cost-Effective Live Cell Structured Illumination Microscopy with Video-Rate Imaging. *ACS Photonics* **8**, 1639–1648 (2021).
19. Dan, D. *et al.* DMD-based LED-illumination Super-resolution and optical sectioning microscopy. *Sci Rep* **3**, 1116 (2013).
20. Li, M. *et al.* Structured illumination microscopy using digital micromirror device and coherent light source. in *Optics in Health Care and Biomedical Optics X* vol. 11553 1155313 (SPIE, 2020).
21. Aydın, M. *et al.* An LED-Based structured illumination microscope using a digital micromirror device and GPU accelerated image reconstruction. *PLOS ONE* **17**, e0273990 (2022).
22. Křížek, P., Raška, I. & Hagen, G. M. Flexible structured illumination microscope with a programmable illumination array. *Opt. Express* **20**, 24585 (2012).

REVIEWERS' COMMENTS

Reviewer #1 (Remarks to the Author):

The authors have comprehensively addressed all points raised by me. And after reading the authors' responses to the other two reviewers, I think they are also reasonable and adequate. I am happy to recommend publication.

Xuesong Li

Reviewer #2 (Remarks to the Author):

Many thanks for the detailed processing of the comments and points of criticism that arose in the previous review. The manuscript has made a significant leap in quality thanks to the thorough editing. In particular, the points raised by Reviewer #1 regarding the theoretical treatment and the subsequent discussion in the document contribute to a significant increase in quality.

With the update of the documentation on Notion, the project can be used as a prime example of good open-source documentation. The authors have gone to great lengths to ensure that others can reproduce the project. With this information, other developers should also be able to migrate the existing software to e.g. Micro-Manager/Pycromanager or a complete Python-based framework.

Since parts of the software, the documentation and assembly guides are now spread along Zenodo, Github and Notion, perhaps the authors can have one central spot (i.e. landing page), where users see the tree of different information at first sight.

After going through all the changes and reviewing the detailed documentation, as well as processing the now available raw data with our own algorithms and installing the Labviewer routine, I support a publication in Nature Communications.

Reviewer #3 (Remarks to the Author):

I have carefully read the author's response to my own comments as well as to the comments and concerns raised by the other reviewers. I have also read the revised manuscripts. The authors have done an excellent job addressing all concerns and I deem the revised manuscript worthy of publication in Nature Communications, if the authors can add one more point to their manuscript: After reading reviewer 1's comments it is pretty clear that reviewer 1 was misled by the original manuscript. It seems that reviewer 1 thought that this manuscript describes an interference-based super-resolution SIM microscope, which is not the case. The system is, indeed, based on a pattern-projected based approach as the authors have stated nicely and quite well in their response to reviewer 1. However, just explaining this fact might not be sufficient to not get other readers confused. The main reason for this is that the acronym SIM is being used in a large number of applications ranging from super-resolution microscopy all the way to determining object height of objects on a conveyor belt. There is a better term for what the authors are describing here: optical sectioning SIM, or OS-SIM. I think if the authors added the simple two more letters to their description and listed 1-2 original OS-SIM papers as references, then most of the current confusion should be lifted.

Response to Referees Letter

Reviewer #1:

The authors have comprehensively addressed all points raised by me. And after reading the authors' responses to the other two reviewers, I think they are also reasonable and adequate. I am happy to recommend publication.

Reviewer #2:

Many thanks for the detailed processing of the comments and points of criticism that arose in the previous review. The manuscript has made a significant leap in quality thanks to the thorough editing. In particular, the points raised by Reviewer #1 regarding the theoretical treatment and the subsequent discussion in the document contribute to a significant increase in quality. With the update of the documentation on Notion, the project can be used as a prime example of good open-source documentation. The authors have gone to great lengths to ensure that others can reproduce the project. With this information, other developers should also be able to migrate the existing software to e.g. Micro-Manager/Pycromanager or a complete Python-based framework. Since parts of the software, the documentation and assembly guides are now spread along Zenodo, Github and Notion, perhaps the authors can have one central spot (i.e. landing page), where users see the tree of different information at first sight.

We thank the reviewer for raising this point. We restructured the first page of the openSIM webpage so that it serves as a central point of the documentation which guides the user to the individual parts of the documentation on the different platforms.

After going through all the changes and reviewing the detailed documentation, as well as processing the now available raw data with our own algorithms and installing the Labviewer routine, I support a publication in Nature Communications.

Reviewer #3:

I have carefully read the author's response to my own comments as well as to the comments and concerns raised by the other reviewers. I have also read the revised manuscripts. The authors have done an excellent job addressing all concerns and I deem the revised manuscript worthy of publication in Nature Communications, if the authors can add one more point to their manuscript: After reading reviewer 1's comments it is pretty clear that reviewer 1 was misled by the original manuscript. It seems that reviewer 1 thought that this manuscript describes an interference-based super-resolution SIM microscope, which is not the case. The system is, indeed, based on a pattern-projected based approach as the authors have stated nicely and quite well in their response to reviewer 1. However, just explaining this fact might not be sufficient to not get other readers confused. The main reason for this is that the acronym SIM is being used in a large number of applications ranging from super-resolution microscopy all the way to determining object height of objects on a conveyor belt. There is a better term for what the authors are describing here: optical sectioning SIM, or OS-SIM. I think if the authors added the simple two more letters to their description and listed 1-2 original OS-SIM papers as references, then most of the current confusion should be lifted.

We thank the reviewer for the suggestion and agree that the term SIM finds application in a multitude of contexts which can lead to ambiguity. We would like to clarify that the openSIM system does not solely provide optical sectioning capabilities but also an improved lateral resolution. Thereby, it can be considered as SR-SIM system in our perspective. OS-SIM systems normally utilize illumination patterns that are relatively coarse¹ whereas SR-SIM systems use illumination pattern with a high spatial frequency². The open-source program (SIMToolbox) we are using to process the SIM data provides OS-SIM and SR-SIM processing methods. Since an LCOS microdisplay is used which

allows to display illumination patterns with arbitrary spatial frequencies in combination with the option to apply different processing algorithms, openSIM offers both OS-SIM but also SR-SIM capabilities. For the data we are presenting in the manuscript we were using illumination patterns with a high spatial frequency and a SR-SIM reconstruction algorithm. We indeed did not mention this aspect in the methods part. In order to place our system in this context and to improve the clarity we incorporated additional information about OS-SIM and SR-SIM, with respective relevant references and added more details about the used spatial frequency and processing algorithm for the presented images.

Manuscript methods section 'Illumination pattern': The chosen illumination patterns had a high spatial frequency.

Manuscript methods section 'Optical design openSIM': Depending on the optical configuration, the spatial frequency of the illumination patterns employed and the reconstruction algorithm, SIM can be used for optical sectioning by reducing out of focus light (OS-SIM)^{3,4} and for super-resolution imaging by enhancing the resolution^{5,6}. For OS-SIM relatively coarse illumination patterns are used¹, while SR-SIM utilizes illumination patterns with a high spatial frequency close to the diffraction limit². LCOS microdisplays provide the flexibility to display illumination patterns with arbitrary spatial frequencies.

Manuscript methods section 'Image processing': The program provides OS-SIM and SR-SIM processing algorithms. For the presented images, the SR-SIM reconstruction method was used.